# Oxidative Stress in the Regulation of Autosis-Related Proteins

**DOI:** 10.3390/antiox14080958

**Published:** 2025-08-04

**Authors:** María Guerra-Andrés, Inés Martínez-Rojo, Alejandra Piedra-Macías, Elena Lavado-Fernández, Marina García-Macia, Álvaro F. Fernández

**Affiliations:** 1Departamento de Bioquímica y Biología Molecular, Universidad de Oviedo, 33006 Oviedo, Spain; guerraamaria@uniovi.es (M.G.-A.); uo277802@uniovi.es (A.P.-M.); 2Instituto Universitario de Oncología del Principado de Asturias (IUOPA), 33006 Oviedo, Spain; 3Instituto de Biología Funcional y Genómica (IBFG), Universidad de Salamanca/CSIC, 37007 Salamanca, Spain; inesmartinez21@usal.es (I.M.-R.); ellavadof@usal.es (E.L.-F.); 4Departamento de Bioquímica y Biología Molecular, Universidad de Salamanca, 37007 Salamanca, Spain; 5Instituto de Investigación Sanitaria del Principado de Asturias (ISPA), 33011 Oviedo, Spain

**Keywords:** autosis, ROS, oxidative stress, Beclin-1, Na^+^,K^+^-ATPase, autophagy

## Abstract

Physiological levels of reactive oxygen species (ROS) play a crucial role as intracellular signaling molecules, helping to maintain cellular homeostasis. However, when ROS accumulate excessively, they become toxic to cells, leading to damage to lipids, proteins, and DNA. This oxidative stress can impair cellular function and lead to various forms of cell death, including apoptosis, necroptosis, ferroptosis, pyroptosis, paraptosis, parthanatos, and oxeiptosis. Despite their significance, the role of ROS in autosis (an autophagy-dependent form of cell death) remains largely unexplored. In this review, we gather current knowledge on autotic cell death and summarize how oxidative stress influences the activity of Beclin-1 and the Na^+^,K^+^-ATPase pump, both of which are critical effectors of this pathway. Finally, we discuss the theoretical potential for ROS to modulate this type of cell death, proposing a possible dual role for these species in autosis regulation through effectors such as HIF-1α, TFEB, or the FOXO family, and highlighting the need to experimentally address cellular redox status when working on autotic cell death.

## 1. Introduction

The oxidoreduction (redox) state of a cell is the center of its metabolism. Almost all metabolic pathways include redox reactions, which affect the redox state of the cell and, at the same time, are influenced by it [1]. The redox state of cells is primarily established by the ratios of the cofactors NADH/NAD^+^, NADPH/NADP^+^, and glutathione (GSH)/glutathione disulfide (GSSG), as well as the balance between reactive oxygen species (ROS) and antioxidants [2]. Maintaining these ratios is essential for cellular homeostasis, which is critical for its correct function and adaptation to environmental stressors. Interestingly, moderate levels of ROS are beneficial, as they preserve, modulate, and regulate cellular functions. However, excessive production of ROS causes oxidative stress, which provokes cellular damage and dysfunction, affecting different cellular mechanisms, including autophagy [3].

Autophagy is an evolutionarily well-conserved pathway essential for cell survival, characterized by the delivery of intracellular components to the lysosome for degradation and recycling. To date, three major forms of autophagy have been described: macroautophagy, microautophagy, and chaperone-mediated autophagy [4]. Amongst them, macroautophagy remains the most studied autophagy variant, and it is defined by the formation of double-membraned vesicles (termed “autophagosomes”) that sequester cellular cargo. These autophagosomes eventually fuse with lysosomes, where pH-sensitive hydrolases mediate degradation of the enclosed material [5]. Autophagy is essential to maintain cellular homeostasis and viability, as it avoids the accumulation of damaged intracellular components and ensures the metabolic needs of the cell during stress and nutrient starvation conditions [6]. Furthermore, it also plays an important role in additional processes, such as intercellular communication, modulation of immune cell functions, and maintenance of tissue barrier integrity [4]. However, autophagy dysregulation or excessive autophagic flux can lead to autophagy-mediated cell death, as it occurs with autosis [7].

In this review, we summarize our current knowledge on autotic cell death and how oxidative stress influences the activity of Beclin-1 and the Na^+^,K^+^-ATPase pump, crucial mediators of autosis. Finally, we discuss the potential for ROS to modulate this type of cell death, hypothesizing a possible dual role for these species in autosis regulation and highlighting the need to address redox status when working on autotic cell death.

## 2. ROS and Oxidative Stress

Oxidative stress is a cellular phenomenon characterized by an imbalance between the production of ROS and the antioxidant defence mechanisms. Under normal physiological conditions, the presence of antioxidants neutralizes these reactive species, maintaining cellular homeostasis. However, some factors, such as radiation (e.g., UV), inflammation, and different metabolic processes, can disrupt this balance and lead to excessive ROS production [8]. These species are highly reactive and can cause damage to cellular components such as DNA, proteins, and lipids. These aggressions can lead to mutations in critical genes, alterations in signaling pathways, and impaired cellular functions [9,10]. For instance, the accumulation of DNA damage induced by ROS in cancer can contribute to genetic instability and the development of malignant tumors, promoting proliferation, angiogenesis, and resistance to cell death, providing favorable conditions for tumor growth and metastasis [11,12]. In addition to ROS, reactive nitrogen species (RNS) also play a significant role in oxidative stress, contributing to cellular and DNA damage and affecting mitochondrial functions [13].

Mitochondria are the primary intracellular source of reactive oxygen species (ROS). During oxidative phosphorylation, complexes I and III of the mitochondrial electron transport chain generate superoxide anions (O_2_^•—^) as byproducts of electron leakage. These anions can be converted into hydrogen peroxide (H_2_O_2_), which, in the presence of transition metal ions such as Fe^2+^ or Cu^+^, leads to the formation of highly reactive hydroxyl radicals (^•^OH) through the Fenton reaction [14]. In addition to this canonical pathway, mitochondria can also produce ROS via reverse electron transport (RET), a process in which electrons flow backwards from complex II to complex I. This phenomenon occurs under specific metabolic conditions, such as high substrate availability and elevated mitochondrial membrane potential, and results in a burst of superoxide production. RET has gained attention as a potent and regulated source of mitochondrial ROS, with implications in both physiological signaling and pathological oxidative damage [15]. Beyond mitochondria, several non-mitochondrial sources contribute to the intracellular generation of reactive oxygen species. One of the most prominent is NADPH oxidase (NOX), a key enzyme involved in inflammatory responses, particularly in immune cells such as neutrophils and macrophages, where it produces superoxide during the respiratory burst [16]. Furthermore, xanthine oxidase, involved in purine metabolism, also generates superoxide as a byproduct of its enzymatic activity [17]. Under pathological conditions, nitric oxide synthase (NOS) can shift from its normal function and contribute to oxidative stress by producing peroxynitrite (ONOO—) through the reaction of nitric oxide with superoxide [18]. Additionally, enzymes in the endoplasmic reticulum, such as members of the cytochrome P450 family, can generate ROS during the metabolism of xenobiotics, especially when ER stress or substrate overload occurs [19]. While all of these systems play essential roles in normal cellular physiology, their dysregulation can lead to excessive ROS accumulation and significant oxidative damage.

Oxidative stress has been related to many pathological conditions, such as cancer, cardiovascular diseases, diabetes, and neurodegenerative diseases [20,21,22,23]. In fact, it has been identified as a key mediator of cell death processes in several of these diseases. For example, oxidative stress can contribute to the emergence of ferroptosis cell death in some cardiovascular diseases, such as atherosclerosis or ischemia [24,25]. In addition, several cell death mechanisms that have been demonstrated to be related to oxidative stress, like apoptosis, necroptosis, or autophagy-dependent cell death, are also relevant in neurodegenerative diseases [26,27,28]. Cells activate different pathways when they detect ROS accumulation to prevent further damage, namely NF-κB, MAPK/p38, or JNK (c-Jun N-terminal kinase), which may induce apoptosis if necessary [29]. A less dramatic response can be mediated by autophagic activation, which would be induced by different pro-survival mediators, such as AMPK or nuclear factor erythroid 2-related factor 2 (NRF2), or directly activating autophagic proteins like ATG4 [30,31,32]. But autophagy can also result in cellular death when exacerbated [33], which could happen in response to uncontrolled oxidative stress. Thus, it is necessary to evaluate the possible interplay between ROS and autophagic cell death and clarify how these species regulate autophagic cell death modes such as autosis.

## 3. Autotic Cell Death

Autophagy is a conserved catabolic pathway that mediates the lysosomal degradation of intracellular threats and long-lived organelles, enabling cells to adapt to environmental stresses such as nutrient deprivation or hypoxia [34]. Even though autophagy has been mainly described as a pro-survival cellular mechanism, it was linked to cell death long ago. Historically, the classification of cell death has been based on morphological traits, distinguishing type I (apoptosis), type II (autophagic cell death), and type III (necrosis) [35]. Accordingly, the term “autophagic cell death” was first used upon ultrastructural features such as the accumulation of autophagosomes. However, autophagy can sometimes be induced during cell death as a protective mechanism, without being responsible for the final demise of the cell. For this reason, the Nomenclature Committee on Cell Death (NCCD) recommendation is to use the term “autophagic cell death” only when the causative role of autophagy has been proven, showing that it can be blocked by genetic or chemical inhibition of at least two components of the autophagic machinery [33] (Figure 1).

Autosis, a new form of cell demise that differs from “classical” autophagic cell death (Table 1), was originally described in 2013 following the aforementioned criteria. Moreover, pharmacological and genetic approaches demonstrated its high dependence on the Na^+^,K^+^-ATPase pump and the autophagy machinery. However, it is important to note that autosis was only inhibited when early stages of autophagy (but not autophagosome–lysosome fusion) were blocked. Interestingly, no autosis reduction was detected when either apoptosis or necroptosis was repressed, showing its independence from other types of cell death [7]. As previously mentioned, the biochemical signature of this process is its dependence on Na^+^/K^+^-ATPase. Accordingly, a large chemical screening identified cardiac glycosides, known natural antagonists of the pump, as potent inhibitors of autosis [7,36]. Further studies, such as a genome-wide siRNA screening, confirmed the implication of Na^+^,K^+^-ATPase in autosis and identified potential new drivers that need to be investigated in the future. Additionally, these studies have also shown that the physical interaction between the pump and Beclin-1, a key autophagy protein, plays a crucial role in autosis [37,38].

From a morphological point of view, autosis can be divided into three phases, characterized by distinctive ultrastructural changes (Figure 2). In what has been called phase 1a, a gradual increase in vacuolar dynamics is observed, with autophagosome accumulation, endoplasmic reticulum dilation and fragmentation, and changes in mitochondria morphology. Later on, during phase 1b, the external and internal membranes separate, with this newly formed perinuclear space occasionally containing inclusions with density and granularity resembling the cytosol. Finally, phase 2 can be described as a rapid and abrupt collapse of the cell, characterized by the shrinkage of the nucleus, which acquires a concave morphology due to the ballooning of the perinuclear space where the nuclear membranes show their maximal separation [7]. Besides these unique features, the dying cell resembles a necrotic one, with swollen mitochondria, absence of the endoplasmic reticulum, and final rupture of the plasma membrane [7,36]. Interestingly, cells become more adherent during autosis and remain on the plate (not floating away) when growing in vitro. Even though light microscopy allows the observation of some of these features, electron microscopy remains the gold standard for the identification of autotic cells (for example, to correctly identify the separation between nuclear membranes) [39].

The first observation of this type of death occurred in HeLa cells exposed to increasing doses of autophagy-inducing peptides such as Tat-beclin-1 and Tat-vFLIP α2, showing a time-dose dependent induction of cell death [7]. Additionally, this process has also been observed in other contexts, such as in vitro nutrient starvation and hypoxia, or different models of ischemia [7,37,40,41]. Interestingly, autosis has also been identified in the livers of patients with severe anorexia nervosa [42], as well as different immune and cancer cells [43,44,45,46,47]. Even in other vertebrates, like zebrafish, features similar to autosis have also been observed; however, they were not defined as such at that moment [48].

## 4. Oxidative Stress and Autosis

Even though it was first reported that antioxidants that block ROS-mediated cell death cannot repress autosis [7], recent reports have shown that the activity of essential autotic proteins (mainly, Beclin-1 and the Na^+^,K^+^-ATPase pump) can be altered by oxidative stress, as it is explained in the next subsections.

### 4.1. Beclin-1 and Oxidative Stress 

Beclin-1 is an essential protein required for the formation of autophagosomes during macroautophagy. This process is initiated when the ULK1 complex, formed by ULK1/ATG13/ATG101/RB1CC1 (FIP200) proteins, is activated in membranes containing ATG9 [49,50,51]. ULK1, acting with the protein SRC, phosphorylates ATG9, promoting the translocation of ATG9-containing vesicles to the autophagy initiation sites [52]. This allows the incorporation of phospholipids from sources such as the endoplasmic reticulum (ER), recycling endosomes, and mitochondria, initiating the elongation of pre-autophagosomal membranes [53]. A complex with Class III phosphatidylinositol 3-kinase (PI3K) activity, containing Beclin-1/PK3C3 (VPS34)/PI3R4 (VPS15) proteins [54,55], is then recruited to these membranes by ATG14 and NRBF2 [56,57], resulting in the production of phosphatidylinositol 3-phosphate (PI3P) by PK3C3 (VPS34). The action of members of the WIPI family and the union of PI3P-binding proteins to these autophagosomal membranes sustain their expansion until their closure [58].

Recent studies have described that Beclin-1 modulation may influence cellular response towards oxidative stress. For instance, Guo et al. demonstrated that with prolonged glucose starvation and hypoxia, ROS levels increase and activate the ATM/CHK2/Beclin-1 axis, promoting autophagy to control excessive ROS accumulation, clear damaged mitochondria, and inhibit apoptosis. During this process, CHK2 phosphorylates Beclin-1 at Ser90/93 residues, which are located in the Bcl-2 binding domain. Thus, this phosphorylation disrupts the interaction between Beclin-1 and Bcl-2, promoting autophagy under oxidative stress [59]. In hepatocytes, the inhibition of PSMD14 (RPN11) deubiquitinase protects against hepatic steatosis and insulin resistance induced by a high-fat diet. This is caused, in part, by the stabilization of Beclin-1, supporting autophagy and decreasing oxidative stress in hepatic cells [60]. Another study shows that a sustained period of oxidative stress in the mammary glands of dairy cows suffering from ketosis is a major cause of injury during early lactation. The presence of oxidative stress is attributed to the supraphysiological circulating concentrations of non-esterified fatty acids (NEFA), which also enhance autophagy as a counteracting response [61]. Moreover, silencing of Beclin-1 attenuated autophagy activity and increased the levels of ROS, which once again suggests that Beclin-1 plays an important role against oxidative stress [62]. However, it has also been proven that ROS can inhibit autophagy initiation, specifically repressing Beclin-1 activity. For example, they can trigger TRPM2-dependent Ca^2+^ influx, which activates CAMK2 at both phosphorylation and oxidation levels, and subsequently phosphorylates Beclin-1 on Ser295. This phosphorylation, in turn, decreased the association of Beclin-1 with PK3C3 (VPS34) while increasing its interaction with BCL2, and thus inhibiting autophagy [63].

The relationship between Beclin-1 and oxidative stress is already being explored to develop therapeutic strategies against some diseases, such as cancer. An example is the antineoplastic agent cannabidiol, which promotes the dissociation of Beclin-1 and BCL2 (enhancing its interaction with PK3C3 (VPS34)) through ROS-induced autophagy [64,65].

### 4.2. Na^+^,K^+^-ATPase and Oxidative Stress

Na^+^,K^+^-ATPase is a fundamental enzyme for the maintenance of ionic homeostasis in cells and the regulation of the membrane potential [66]. This pump is a multimeric complex consisting of three subunits: α, β, and γ [67]. The α-subunit is the catalytic component, responsible for ATP hydrolysis and ion transport, while the β-subunit has no catalytic activity but regulates the enzymatic function of the pump and confers stability to the α-subunit [68]. There are four isoforms of the α-subunit in humans: α1 is the dominant isoform and is widely expressed in almost all cell types [66], α2-isoform is predominantly produced in both cardiac and skeletal muscle as well as in the brain (in astrocytes and glia cells) [66,69], α3 is mainly located in neurons and cardiac cells (with gender-specific differences in the latter) [70,71], and α4-isoform is only expressed in the testes, where it has been associated with sperm motility [72]. There are also three isoforms of the β-subunit [73]: β1 is found in most tissues [66], while β2 is expressed in the colon, neurons, and cardiac cells [69,74,75], and β3 is detected in the testes [76]. Finally, mammals express up to seven isoforms of the γ-subunit (also known as the FXYD family), and they are responsible for the regulation of the affinity of the enzyme to different ligands [77]. Besides modulating the ionic gradient of Na^+^ and K^+^, the pump also forms complexes with proteins from the plasma membrane, like caveolin, SRC, EGFR, or GPR35, which enable it to participate in different signaling pathways [78,79]. Furthermore, the pump can also modulate adhesion and migration between cells [80].

Many stimuli induce specific modifications in Na^+^,K^+^-ATPase and can change its activity [81,82], ROS being one of them. For example, Na^+^,K^+^-ATPase α1-subunit has been observed to be degraded during hypoxia-induced pulmonary edema due to mitochondria-generated ROS and the participation of the ubiquitin-conjugating system [83]. During some neurodegenerative diseases, such as Alzheimer’s, trans fatty acids like linoelaidic acid enhance oxidative stress and decrease Na^+^,K^+^-ATPase activity. Furthermore, it highly reacts with amyloid β (Aβ) depositions, causing even more severe oxidative damage [84]. Also, long-term exposure to Aβ can change the thiol redox status of human neuroblastoma cells, inhibiting the pump and leading to the induction of glutathionylation of the α-subunit [85]. S-glutathionylation is, in fact, one of the most documented oxidative modifications of Na^+^,K^+^-ATPase. In the case of cardiac sarcolemma, the activation of the renin–angiotensin system also leads to the glutathionylation of the β1 subunit of the pump, causing its inhibition [86]. Similarly, oxidative stress from placental ischemia/reperfusion and hypoxia in preeclampsia also causes S-glutathionylation of the β1 subunit, hampering its function [87]. S-glutathionylation of cysteine residues of the α1 and β1 subunits can also reduce the α1/β1 association, causing conformational changes and blocking the α1 subunit’s intracellular ATP-binding site, leading to the inhibition of its activity [88,89]. Nevertheless, the oxidative modifications of Na^+^,K^+^-ATPase appear to be reversible [90,91], which means its activity can be regulated in a redox-sensitive manner. Furthermore, members of the FXYD family can also reverse oxidative stress-induced inhibition of the pump activity by deglutathionylation of the β1 subunit [91]. Still, the exposure of the pump to free radicals increases its susceptibility to degradation by proteolytic enzymes [92,93].

Apart from ROS, RNS can also modify Na^+^,K^+^-ATPase activity. Peroxynitrite, for example, is produced when nitric oxide and superoxide react [94], and it can act as a potent irreversible inhibitor of the pump activity by both nitrating tyrosine residues in all three subunits and modifying cysteine residues [95]. Furthermore, the free radical nitric oxide (^•^NO) can also regulate Na^+^,K^+^-ATPase through the activation of soluble guanylate cyclase and cGMP in the central nervous system, modulating cerebral blood flow and synaptic transmission [96,97]. The regulation of the pump activity by ^•^NO is also known in other tissues, such as the choroid plexus [98].

Interestingly, oxidative stress can also modulate the interaction of the pump with other proteins, like the Na^+^,K^+^-ATPase/SRC axis. The α1-subunit serves as a scaffold for proto-oncogene tyrosine kinase c-SRC, interacting with it and inhibiting its function [99]. However, when cardiac glucosides bind to the extracellular domain of the pump, a conformational change is produced, allowing SRC to be activated. In turn, it transactivates EGFR, which then activates RAS [100]. This can lead to the production of ROS by the mitochondria, resulting in the activation of nuclear factor kappa B (NF-κB) and initiating the MAPK cascade [101,102]. Furthermore, SRC may also be regulated by ROS through a reversible oxidation of cysteine residues, generating a positive feedback loop [90]. Some of these oxidized residues are located where the interaction between Na^+^,K^+^-ATPase and SRC takes place [99], inducing changes in either the pump, SRC, or both, which in turn might affect the interaction between them and, consequently, the downstream kinase cascade [103].

## 5. A New Layer in Autosis Regulation

A decade has passed since autosis was first described, yet our understanding of this specific type of cell death remains limited. While the Na^+^,K^+^-ATPase pump and certain components of the autophagy machinery are crucial for initiating this cellular response, the complete molecular pathway and its regulatory mechanisms are still largely unexplored. Most research has concentrated on potential ionic imbalances caused by dysregulated activity of the pump, as well as oxygen and nutrient deprivation, since both hypoxia and starvation can trigger autosis. However, in this review, we have displayed recent studies that indicate that ROS can also affect the function of autotic components. Therefore, we propose that the redox status of the cell should be considered when investigating the molecular and regulatory mechanisms of autosis.

In this regard, it is noteworthy that oxidative stress can inhibit both Beclin-1 and Na^+^,K^+^-ATPase [63,88,104,105,106,107,108], for example, by modifications such as S-glutathionylation, tyrosine nitration, and others [89,95,109]. This suggests that these species may actually prevent autotic induction at first, rather than promote it, as the repression of Beclin-1 and pump activities can effectively block autosis [7]. Depending on the context, ROS can act as important signaling molecules instead of toxins, helping to maintain cellular homeostasis by allowing cells to rapidly respond to various conditions for survival. In this sense, ROS may first negatively affect Beclin-1 and the pump, reducing their activity, preventing their binding, and hindering the autotic response. However, an overwhelming increase in ROS production could concurrently promote an exacerbated autophagic response that is triggered as an attempt to counteract oxidative stress [110], but that can ultimately lead to the initiation of autosis. This could be particularly relevant, for example, under hypoxic conditions, which are associated with both elevated ROS generation [111,112] and autotic cell death [7,37,38], potentially through HIF-1α. This is an essential transcription factor that can be activated and stabilized in response to hypoxia-induced ROS [113], resulting in the promotion of autophagy [114,115]. Another transcription factor that could be important in this process is TFEB, which is also activated under oxidative stress [116,117] and promotes the expression of autophagic and lysosomal genes [118]. Other interesting effectors that can activate autophagy in response to ROS (even under hypoxic conditions) are components of the FOXO family, such as FOXO1 and FOXO3 [119,120,121,122,123]. But how much oxidative stress would be needed to activate autosis? What is the threshold for this switch in the ROS-autosis interplay? These are complex questions that would need to be experimentally addressed. Nevertheless, it seems it could greatly depend on the context. For example, several studies have described that AMPK, one of the main autophagy regulators, can be both repressed and stimulated by oxidative stress, depending on whether ROS mediates the phosphorylation (positive regulation) or the oxidation (negative regulation) of its residues [124].

In conclusion, based on our current understanding of ROS and autotic-related proteins, we suggest that these species could have a dual role in autosis (Figure 3). First, ROS may function as molecular brakes for this type of cell death by inhibiting the activity of Beclin-1 and the Na^+^,K^+^-ATPase pump. However, when present in high concentrations, ROS can activate different autophagy-inducing signaling pathways (through mediators such as HIF-1α, TFEB, FOXO, or AMPK), exacerbating the autophagy response and leading to autosis initiation. Nevertheless, this conceptual model must be experimentally confirmed in the future to fully unveil the role of ROS and oxidative stress in this type of autophagy-dependent cell death. Therefore, we believe it is crucial to consider the redox status of the cell when examining the molecular and regulatory mechanisms of autosis, as it can influence the function of key effectors in the pathway.

## Figures and Tables

**Figure 1 antioxidants-14-00958-f001:**
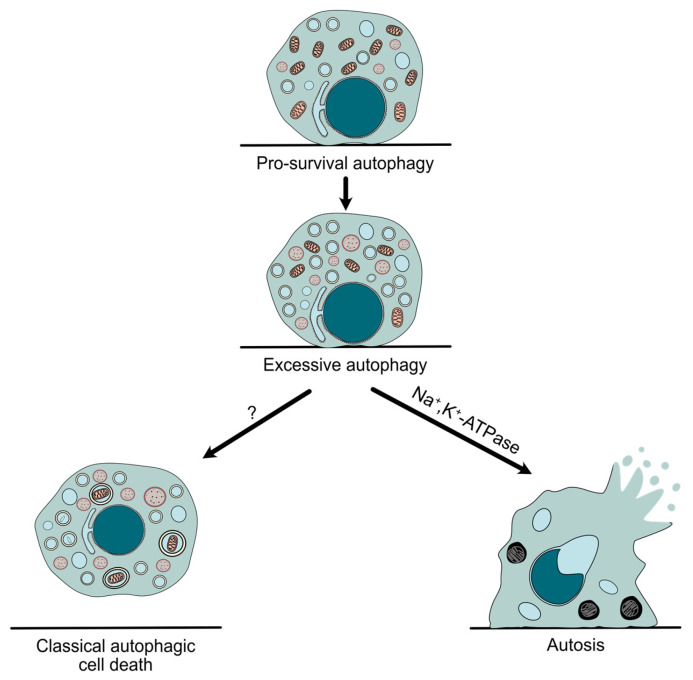
**Autophagy in cell survival and cell death.** Although autophagy often acts as a protective mechanism, its dysregulation can lead to an exacerbated response that results in autophagy-mediated cell death, such as autosis, which differs from other possible types of autophagic cell death due to its unique morphological features, as well as its dependence on Na^+^,K^+^-ATPase.

**Figure 2 antioxidants-14-00958-f002:**
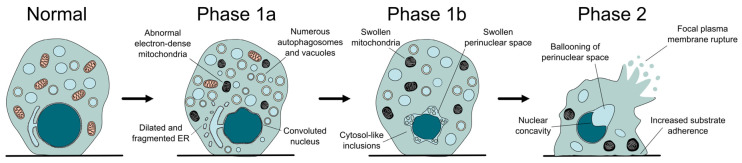
**The different phases of autotic cell death.** During autosis, the cell undergoes several morphological changes, some of which are distinctive of this process, like the separation of the nuclear membranes, the increased attachment to the substrate, or the formation of a balloon and a concavity in the surface of the nucleus.

**Figure 3 antioxidants-14-00958-f003:**
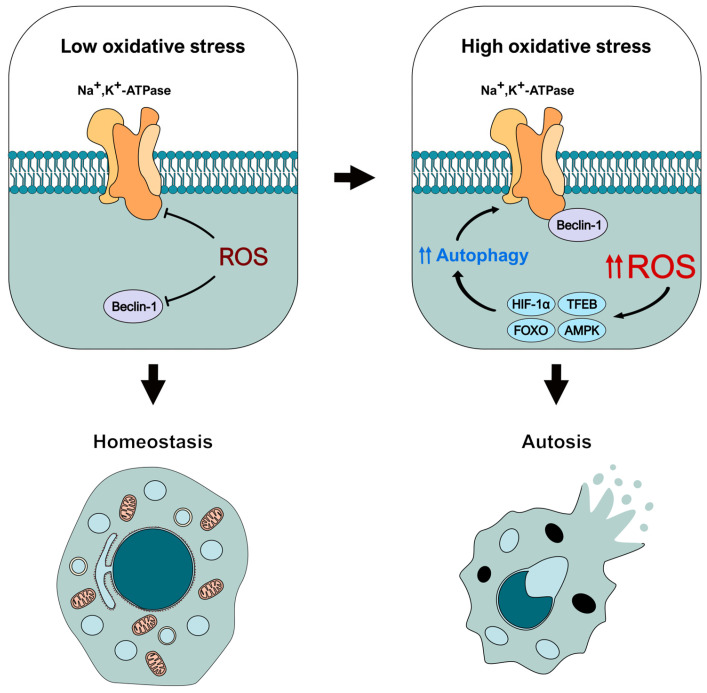
**Theoretical model of the dual regulation of autosis-related effectors by oxidative stress.** ROS would block autosis initiation early by inhibiting the activity of Beclin-1 and Na^+^,K^+^-ATPase. However, increased oxidative stress would trigger an intense autophagic response mediated by different effectors (such as HIF-1 α, TFEB, or AMPK) that could lead to autosis initiation.

**Table 1 antioxidants-14-00958-t001:** Morphological differences between autophagy-dependent cell death and autosis.

Cellular Structure	“Classical” Autophagy-Dependent Cell Death	Autosis
Nucleus	Minor changes	Nuclear membrane convolution Detachment of inner and outer nuclear membranes Swollen perinuclear space and focal ballooning Focal concavity of the inner nuclear surface
Chromatin	Minor changes	Chromatin condensation
Plasma membrane	Plasma membrane rupture Occasional blebbing	Focal plasma membrane rupture
Autophagic structures	Numerous autophagic vesicles	Numerous autophagic vesicles in early stages
Organelles	Occasional ER, mitochondria and Golgi enlargement Occasional depletion of organelles	Electrodense and swollen mitochondria ER dilation, fragmentation (early stages) and disappearance (late stages)
Other features		Cell-substrate adhesion Na^+^,K^+^-ATPase dependence

## Data Availability

No new data were created or analyzed in this study. Data sharing is not applicable to this article.

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
