# Peer review of "Oxidative Stress in the Regulation of Autosis-Related Proteins"

_antioxidants, 2025, doi:10.3390/antiox14080958_

Round 1

Reviewer 1 Report

The manuscript titled "Oxidative stress in the regulation of autosis-related proteins" systematically reviews the role of reactive oxygen species (ROS) in regulating autosis, with a focus on the modulation of Beclin-1 and Na⁺,K⁺-ATPase, and proposes a dual regulatory role of ROS in autosis. While the topic is relevant to cell biology and redox signaling, several critical issues need to be addressed to enhance its scientific rigor and comprehensiveness. 

The manuscript titled "Oxidative stress in the regulation of autosis-related proteins" systematically reviews the role of reactive oxygen species (ROS) in regulating autosis, with a focus on the modulation of Beclin-1 and Na⁺,K⁺-ATPase, and proposes a dual regulatory role of ROS in autosis. While the topic is relevant to cell biology and redox signaling, several critical issues need to be addressed to enhance its scientific rigor and comprehensiveness. 

1. The reference list includes disproportionately few studies published after 2022 (e.g., only 1 in 2023 and 2 in 2024). This gap may lead to overlooking significant advances in autosis mechanisms, ROS-mediated post-translational modifications of key proteins, and novel in vivo models. The authors should comprehensively update the literature to incorporate recent findings, particularly those exploring the crosstalk between redox signaling and autotic pathways.

2. The proposed dual role of ROS in autosis (inhibition at low levels, promotion at high levels) in Section 5 remains largely conceptual. The authors must elaborate on specific molecular events, such as ROS-induced oxidative modifications (e.g., cysteine glutathionylation, tyrosine nitration) of Beclin-1 and Na⁺,K⁺-ATPase, and define the threshold concentrations of ROS that shift their functional outcomes. Mechanistic details (e.g., downstream signaling cascades) should be reinforced with cited evidence.

3.Section 3 emphasizes autosis as a distinct cell death type but neglects ongoing debates, such as its morphological and molecular overlap with other autophagy-dependent cell death modalities. The authors should critically compare autosis with canonical autophagic cell death, addressing discrepancies in defining criteria (e.g., reliance on Na⁺,K⁺-ATPase vs. general autophagy machinery) to provide a balanced perspective. 

4. The transition from Section 2 (ROS and oxidative stress) to Section 3 (Autotic cell death) is abrupt. A bridging paragraph is needed to explicitly link ROS (as central regulators of cellular metabolism) to autosis (a metabolism-sensitive cell death pathway), establishing a clear rationale for their functional interplay. This will enhance the narrative flow and highlight the relevance of redox status in autosis regulation.

5. The dual role conclusion in Section 5 is primarily supported by studies from Guo et al. (2020) and Wang et al. (2016), which are insufficient to validate the hypothesis. The authors must either strengthen the argument with additional literature demonstrating context-dependent ROS effects or temper the claim to reflect its speculative nature, acknowledging the need for further experimental validation.

Major Revision 

The manuscript addresses an important topic but requires substantial improvements in literature coverage, mechanistic depth, balanced discussion, and figure clarity. Addressing these issues will strengthen its contribution to understanding redox regulation of autosis. 

Author Response

We sincerely appreciate your time and effort in reviewing our manuscript titled "Oxidative stress in the regulation of autosis-related proteins". We highly value your comments and suggestions, which have undoubtedly helped improve the clarity and scientific rigour of the manuscript through the changes we have made (marked with red font in the revised manuscript). We would like to clarify that this article is, in fact, a conceptual work aimed at gathering the little information that is available about oxidative stress and the autotic proteins Beclin-1 and Na⁺,K⁺-ATPase and proposing a hypothesis that undoubtedly will need to be experimentally confirmed in the future.  

Please, find below our point-by-point response (in Times New Roman) to all the comments:

Reviewer 1

The manuscript titled "Oxidative stress in the regulation of autosis-related proteins" systematically reviews the role of reactive oxygen species (ROS) in regulating autosis, with a focus on the modulation of Beclin-1 and Na⁺,K⁺-ATPase, and proposes a dual regulatory role of ROS in autosis. While the topic is relevant to cell biology and redox signaling, several critical issues need to be addressed to enhance its scientific rigor and comprehensiveness. 

  1. The reference list includes disproportionately few studies published after 2022 (e.g., only 1 in 2023 and 2 in 2024). This gap may lead to overlooking significant advances in autosis mechanisms, ROS-mediated post-translational modifications of key proteins, and novel in vivo models. The authors should comprehensively update the literature to incorporate recent findings, particularly those exploring the crosstalk between redox signaling and autotic pathways.

Authors’ response: We thank the reviewer for the suggestion and agree that the manuscript should include the most recent studies. Unfortunately, there are very few papers on autosis, all of which have already been cited in the original manuscript. While there are some additional articles that mention the term “autosis,” these either do not provide convincing data, such as a clear separation of nuclear membranes or dependencies on the Na⁺,K⁺-ATPase, or they use the term “autosis” interchangeably with “autophagic cell death.” This confusion remains prevalent in the field. In our manuscript, we have chosen to include only those references that clearly demonstrate autotic cell death.

Nevertheless, we have thoroughly reviewed the manuscript and updated the references with more recent papers where possible.

  1. The proposed dual role of ROS in autosis (inhibition at low levels, promotion at high levels) in Section 5 remains largely conceptual. The authors must elaborate on specific molecular events, such as ROS-induced oxidative modifications (e.g., cysteine glutathionylation, tyrosine nitration) of Beclin-1 and Na⁺,K⁺-ATPase, and define the threshold concentrations of ROS that shift their functional outcomes. Mechanistic details (e.g., downstream signaling cascades) should be reinforced with cited evidence.

Authors’ response: We agree with the reviewer that Section 5 is conceptual, as it presents our proposed hypothesis regarding the role of reactive oxygen species (ROS) in the regulation of autosis. To clarify this perspective and provide a stronger foundation, we have rewritten and expanded the section, incorporating additional references that could support our model. We have also mentioned potential downstream signaling cascades, including HIF-1α, TFEB, the FOXO family, and AMPK. Furthermore, we have emphasized the theoretical nature of this section and noted that it requires empirical validation.

  1. Section 3 emphasizes autosis as a distinct cell death type but neglects ongoing debates, such as its morphological and molecular overlap with other autophagy-dependent cell death modalities. The authors should critically compare autosis with canonical autophagic cell death, addressing discrepancies in defining criteria (e.g., reliance on Na⁺,K⁺-ATPase vs. general autophagy machinery) to provide a balanced perspective. 

Authors’ response: We appreciate the reviewer’s feedback and apologize for not providing a clearer explanation. We have now added additional information to better distinguish autosis from other modes of autophagy-dependent cell death. Specifically, we have included two new figures: Figure 1 illustrates the transition from a beneficial autophagic response to detrimental autophagy and autosis, while Figure 2 depicts the various morphological phases of autosis. Additionally, we have included a table that compares the characteristics of “classical” autophagy-related cell death with autosis.

  1. The transition from Section 2 (ROS and oxidative stress) to Section 3 (Autotic cell death) is abrupt. A bridging paragraph is needed to explicitly link ROS (as central regulators of cellular metabolism) to autosis (a metabolism-sensitive cell death pathway), establishing a clear rationale for their functional interplay. This will enhance the narrative flow and highlight the relevance of redox status in autosis regulation.

Authors’ response: This is another great comment, and a much-needed suggestion. We have now included a bridging paragraph commenting on the link between oxidative stress and different types of cell death, thus obtaining a much better transition from ROS to autophagy (and autotic) cell death.

  1. The dual role conclusion in Section 5 is primarily supported by studies from Guo et al. (2020) and Wang et al. (2016), which are insufficient to validate the hypothesis. The authors must either strengthen the argument with additional literature demonstrating context-dependent ROS effects or temper the claim to reflect its speculative nature, acknowledging the need for further experimental validation.

Authors’ response: As explained before, we have now included new references that can better support the theoretical proposal while also tempering down the tone of the paragraph, explicitly acknowledging that an empirical validation is required.

Reviewer 2 Report

The authors submitted a review article of considerable interest that addresses reactive oxygen species (ROS) messenger agents. The review paper focuses on the participation of ROS in autosis-related proteins. Given that autosis can be classified as a type of programmed cell death, which is involved in many pathological and physiological processes, an understanding of autosis is important and timely.

The authors' treatment of the topic encompassed all its important aspects. First, the role of ROS in intracellular signal cascades is discussed. Subsequently, the authors proceed to address the subject of ROS-induced oxidative stress and the repercussions of ROS concentration imbalances on cellular death. The text employs a single illustrative figure to elucidate an association between elevated ROS levels and beclin-1 coupling with Na⁺, K⁺-ATPase. In addition, the paper makes use of a substantial number of relevant references.

I did not find any serious errors.

Author Response

General comments:

We sincerely appreciate your time and effort in reviewing our manuscript titled "Oxidative stress in the regulation of autosis-related proteins". We highly value your comments and suggestions, which have undoubtedly helped improve the clarity and scientific rigour of the manuscript through the changes we have made (marked with red font in the revised manuscript). We would like to clarify that this article is, in fact, a conceptual work aimed at gathering the little information that is available about oxidative stress and the autotic proteins Beclin-1 and Na⁺,K⁺-ATPase and proposing a hypothesis that undoubtedly will need to be experimentally confirmed in the future.  

Please, find below our point-by-point response to the comments:

Reviewer 2

The authors submitted a review article of considerable interest that addresses reactive oxygen species (ROS) messenger agents. The review paper focuses on the participation of ROS in autosis-related proteins. Given that autosis can be classified as a type of programmed cell death, which is involved in many pathological and physiological processes, an understanding of autosis is important and timely.

 The authors' treatment of the topic encompassed all its important aspects. First, the role of ROS in intracellular signal cascades is discussed. Subsequently, the authors proceed to address the subject of ROS-induced oxidative stress and the repercussions of ROS concentration imbalances on cellular death. The text employs a single illustrative figure to elucidate an association between elevated ROS levels and beclin-1 coupling with Na⁺, K⁺-ATPase. In addition, the paper makes use of a substantial number of relevant references.

Authors’ response: We sincerely thank the reviewer for his/her comments and nonetheless hope that the revised is also well-received.

Reviewer 3 Report

The article is interesting and merits publication. However, I have suggestions for improvement:

  • I recommend including a schematic diagram or comparative table to clearly illustrate the differences between autophagy and autosis.
  • What experimental evidence supports the involvement of ROS in autosis? Please elaborate on this point.
  • Description of Figure 1 is referenced in the manuscript, but I suggest providing a detailed legend directly beneath the figure for clarity.
  • Additionally, the following sentence (Even though it was first reported that antioxidants that block ROS-mediated cell 159 death cannot repress autosis [7], recent reports have shown that the activity of essential 160 autotic proteins (mainly, Beclin-1 and the Na+ , K+-ATPase pump) can be altered by oxida- 161 tive stress) should be expanded into several sentences to make the conclusion more comprehensible. It should also include appropriate literature references to support the claims.
  • Finally, the abstract is too general. It should be revised to include specific findings and key points derived from this review.

The article is interesting and merits publication. However, I have suggestions for improvement:

  • I recommend including a schematic diagram or comparative table to clearly illustrate the differences between autophagy and autosis.
  • What experimental evidence supports the involvement of ROS in autosis? Please elaborate on this point.
  • Description of Figure 1 is referenced in the manuscript, but I suggest providing a detailed legend directly beneath the figure for clarity.
  • Additionally, the following sentence (Even though it was first reported that antioxidants that block ROS-mediated cell 159 death cannot repress autosis [7], recent reports have shown that the activity of essential 160 autotic proteins (mainly, Beclin-1 and the Na+ , K+-ATPase pump) can be altered by oxida- 161 tive stress) should be expanded into several sentences to make the conclusion more comprehensible. It should also include appropriate literature references to support the claims.
  • Finally, the abstract is too general. It should be revised to include specific findings and key points derived from this review.

Author Response

General comments:

We sincerely appreciate your time and effort in reviewing our manuscript titled "Oxidative stress in the regulation of autosis-related proteins". We highly value your comments and suggestions, which have undoubtedly helped improve the clarity and scientific rigour of the manuscript through the changes we have made (marked with red font in the revised manuscript). We would like to clarify that this article is, in fact, a conceptual work aimed at gathering the little information that is available about oxidative stress and the autotic proteins Beclin-1 and Na⁺,K⁺-ATPase and proposing a hypothesis that undoubtedly will need to be experimentally confirmed in the future.  

Please, find below our point-by-point response to all the comments:

Reviewer 3

The article is interesting and merits publication. However, I have suggestions for improvement:

  • I recommend including a schematic diagram or comparative table to clearly illustrate the differences between autophagy and autosis.

Authors’ response: We are very thankful for this recommendation, and we have now included two figures (Figure 1, showing the transition from beneficial autophagic response to detrimental autophagy and autosis; and Figure 2, depicting the different morphological phases of autosis) and a table (that compares the features of “classical” autophagy cell death and autosis) to better differentiate autosis from pro-survival autophagy and other “classical” modes of autophagic cell death.

  • What experimental evidence supports the involvement of ROS in autosis? Please elaborate on this point.

Authors’ response: We appreciate the reviewer’s question, as we recognize the importance of clarifying this subject. Although our proposed hypothesis is currently a theoretical model, there are existing studies that demonstrate ROS-mediated regulation of proteins involved in autosis, such as Beclin-1 and the Na⁺,K⁺-ATPase. This information is presented in Section 4. However, to the best of our knowledge, there is no experimental evidence supporting ROS modulation of autosis. As a result, we have rewritten Section 5 to include additional references that bolster the model, while also softening the tone of the paragraph and clearly stating that empirical validation is needed.

  • Description of Figure 1 is referenced in the manuscript, but I suggest providing a detailed legend directly beneath the figure for clarity.

Authors’ response: We agree with the reviewer that more detailed legends can enhance the paper's readability. Therefore, we have included extended descriptions for both the old and new figures.

  • Additionally, the following sentence (Even though it was first reported that antioxidants that block ROS-mediated cell death cannot repress autosis [7], recent reports have shown that the activity of essential autotic proteins (mainly, Beclin-1 and the Na+ , K+-ATPase pump) can be altered by oxidative stress) should be expanded into several sentences to make the conclusion more comprehensible. It should also include appropriate literature references to support the claims.

Authors’ response: We thank the reviewer for this comment, as it shows that we should have better clarified that this information is fully explained (with references) in the subsections of Section 4. We have now included a new sentence to better explain this matter.

  • Finally, the abstract is too general. It should be revised to include specific findings and key points derived from this review.

Authors’ response: Following the reviewer’s suggestion, we have now rewritten the ending of the abstract to show new information that is included in the revised version of the manuscript, as well as its key points.

Reviewer 4 Report

Comments

The review article “Oxidative stress in the regulation of autosis-related proteins” by Guerra-Andrés et al. has highlighted the role and importance of oxidative stress in the regulation of autosis-associated proteins. The normal level of ROS is important for the regulation of various cellular activities, but an uncontrolled level could harm metabolic and cellular functions. The current review discussed autosis, an autophagy-dependent, non-apoptotic, and non-necrotic form of cell death, marked by features like focal swelling of the perinuclear space, accumulation of autophagic vacuoles, and dependence on Na⁺/K⁺‑ATPase activity. The review is well-written and reports the key findings in the field. I have the following minor comments that need to be addressed in the revised form of the manuscript.

  1. Line 120: includes the original reference that mentioned the autosis.
  2. Line 130: reference missing.
  3. Section 3 requires an illustrative presentation of the autosis pathway. This will help readers to differentiate different forms of cellular cell death events.
  4. Figure 1 needs to be modified to make it more informative. The current image does not explain how autosis gets activated.

Comments

The review article “Oxidative stress in the regulation of autosis-related proteins” by Guerra-Andrés et al. has highlighted the role and importance of oxidative stress in the regulation of autosis-associated proteins. The normal level of ROS is important for the regulation of various cellular activities, but an uncontrolled level could harm metabolic and cellular functions. The current review discussed autosis, an autophagy-dependent, non-apoptotic, and non-necrotic form of cell death, marked by features like focal swelling of the perinuclear space, accumulation of autophagic vacuoles, and dependence on Na⁺/K⁺‑ATPase activity. The review is well-written and reports the key findings in the field. I have the following minor comments that need to be addressed in the revised form of the manuscript.

  1. Line 120: includes the original reference that mentioned the autosis.
  2. Line 130: reference missing.
  3. Section 3 requires an illustrative presentation of the autosis pathway. This will help readers to differentiate different forms of cellular cell death events.
  4. Figure 1 needs to be modified to make it more informative. The current image does not explain how autosis gets activated.

Author Response

General comments:

We sincerely appreciate your time and effort in reviewing our manuscript titled "Oxidative stress in the regulation of autosis-related proteins". We highly value your comments and suggestions, which have undoubtedly helped improve the clarity and scientific rigour of the manuscript through the changes we have made (marked with red font in the revised manuscript). We would like to clarify that this article is, in fact, a conceptual work aimed at gathering the little information that is available about oxidative stress and the autotic proteins Beclin-1 and Na⁺,K⁺-ATPase and proposing a hypothesis that undoubtedly will need to be experimentally confirmed in the future.  

Please, find below our point-by-point response (in Times New Roman) to all the comments:

Reviewer 4

The review article “Oxidative stress in the regulation of autosis-related proteins” by Guerra-Andrés et al. has highlighted the role and importance of oxidative stress in the regulation of autosis-associated proteins. The normal level of ROS is important for the regulation of various cellular activities, but an uncontrolled level could harm metabolic and cellular functions. The current review discussed autosis, an autophagy-dependent, non-apoptotic, and non-necrotic form of cell death, marked by features like focal swelling of the perinuclear space, accumulation of autophagic vacuoles, and dependence on Na⁺/K⁺‑ATPase activity. The review is well-written and reports the key findings in the field. I have the following minor comments that need to be addressed in the revised form of the manuscript.

  • Line 120: includes the original reference that mentioned the autosis.

Authors’ response: We thank the reviewer for this suggestion, and we have now added the aforementioned reference.

  • Line 130: reference missing.

Authors’ response: We would like to apologize to the reviewer, as we have revised the manuscript but could not identify any missing references (the sentence of the indicated line already has a citation). However, we will promptly add any missing citations if further revisions are necessary.

  • Section 3 requires an illustrative presentation of the autosis pathway. This will help readers to differentiate different forms of cellular cell death events.

Authors’ response: We thank the reviewer for this suggestion, and we have now included two figures (Figure 1, showing the transition from beneficial autophagic response to detrimental autophagy and autosis; and Figure 2, depicting the different morphological phases of autosis) and a table (that compares the features of “classical” autophagy cell death and autosis) to better differentiate autosis from pro-survival autophagy and other “classical” modes of autophagic cell death.

  • Figure 1 needs to be modified to make it more informative. The current image does not explain how autosis gets activated.

Authors’ response: The previous Figure 1, now labeled as "Figure 3," has been enhanced to better illustrate the proposed model. This revised version clearly outlines potential mediators, such as HIF-1α, TFEB, the FOXO family, and AMPK, which can induce autophagy in response to increased oxidative stress. Ultimately, this process may lead to autosis by promoting the interaction between Beclin-1 and Na⁺,K⁺-ATPase.

Round 2

Reviewer 1 Report

The authors have adequately addressed all the points raised during the review process. The manuscript has been revised accordingly and now meets the publication standards of *Antioxidants*. I recommend its acceptance and publication in its current form.

The authors have adequately addressed all the points raised during the review process. The manuscript has been revised accordingly and now meets the publication standards of *Antioxidants*. I recommend its acceptance and publication in its current form.

Reviewer 3 Report

  Thank you for your answers. I accept the improvement of the work.

Thank you for your answers. I accept the improvement of the work.